# Reactivity mapping of nanoscale defect chemistry under electrochemical reaction conditions

Jonas H.K. Pfisterer[1], Masoud Baghernejad[1], Giovanni Giuzio [1] & Katrin F. Domke[1]*

Electrocatalysts often show increased conversion at nanoscale chemical or topographic surface inhomogeneities, resulting in spatially heterogeneous reactivity. Identifying reacting species locally with nanometer precision during chemical conversion is one of the biggest quests in electrochemical surface science to advance (electro)catalysis and related fields. Here, we demonstrate that electrochemical tip-enhanced Raman spectroscopy can be used for combined topography and reactivity imaging of electro-active surface sites under reaction conditions. We map the electrochemical oxidation of Au nanodefects, a showcase energy conversion and corrosion reaction, with a chemical spatial sensitivity of about 10 nm. The results indicate the reversible, concurrent formation of spatially separated $Au_2O_3$ and $Au_2O$ species at defect-terrace and protrusion sites on the defect, respectively. Active-site chemical nano-imaging under realistic working conditions is expected to be pivotal in a broad range of disciplines where quasi-atomistic reactivity understanding could enable strategic engineering of active sites to rationally tune (electro)chemical device properties.

[1] Molecular Spectroscopy Department, Electrochemical Surface Science Group, Max Planck Institute for Polymer Research, Ackermannweg 10, 55128 Mainz, Germany. *email: domke@mpip-mainz.mpg.de

**N**anoscale structural or chemical material heterogeneities, or active sites, can markedly define macroscopic material or device properties, like (electro)chemical reactivity and energy conversion[1,2], charge or heat management[3,4] or photonic properties[5]. Controlled active-site engineering thus possesses an enormous potential for the strategic design of functional materials —but requires understanding of surface chemical reactivity at the atomistic or molecular level under operating conditions[6].

One prominent field where active sites play a pivotal role is heterogeneous (electro)catalysis that lies at the basis of many important chemical or energy conversion reactions. Both nature and location of the catalytically active sites are often strongly debated, as atomistic insights obtained from ultrahigh vacuum studies and computer simulations are not easily transferable to catalyst behavior under complex reaction conditions at high pressure and/or temperature or in the presence of electrostatic fields[7,8]. An important stepping stone to close the catalysis discovery loop is the elucidation of defect-defined reaction mechanisms with new operando experimental and theoretical tools[9,10]. Electrochemical scanning tunneling microscopy (EC-STM) has recently been used to identify active surface sites by the comparison of local current fluctuations of the tunneling barrier above reactive and non-reactive nanoscale surface sites[11], but lacks chemical information about the reacting species. Advanced vibrational in situ spectroscopy-like electrochemical shell-isolated nanoparticle-enhanced Raman spectroscopy, on the other hand, has been introduced as an elegant means to monitor reaction intermediate formation at well-defined single crystal electrodes during electrochemical conversion[12,13], albeit with diffraction-limited spatial resolution. Combining EC-STM with the chemical specificity of near-field Raman spectroscopy in the form of electrochemical tip-enhanced Raman spectroscopy (EC-TERS)[14-16] holds great promise to enable the investigation of local active defect-chemistry on the nanometer level during electrocatalytic energy conversion reactions. Using TERS in air, Ren and co-workers have resolved the energetic differences in the step edge and terrace reactivity of Pd islands on Au(111) with adsorbed organic probing molecules reaching a chemical spatial resolution of impressive 3 nm[17]. Kumar et al.[18] recently managed to study the dimerization reaction of *p*-aminothiophenol attached to rough Ag surfaces in contact with a water droplet and the Van Duyne group mapped the spatially heterogeneous surface potential distribution on ITO grains with 40 nm resolution using EC-TERS[19]. The challenge is now to combine the key abilities of these striking works—chemical site-specificity on the nanoscale, 2D in situ reactivity imaging starting from and ending at the plain catalyst surface, and electrochemical (potential) control of reversible catalyst (de)activation—into one single experimental approach.

Here, we demonstrate that EC-TERS can be used to map potential-controlled active-site chemistry with about 10 nm spatial chemical resolution. As a showcase corrosion and heterogeneous catalysis-related reaction[20], we image the oxidation of nanoscale defect protrusions at a Au(111) single crystal electrode generated by electrochemical water splitting at defect sites to form Au oxide. We resolve the correlation between heterogeneous surface topography and electrochemical (re)activity under operando conditions with a spatial precision of ~ 10 nm. Correlating the apparent height of defect structures with the EC-TERS Au oxide (AuOx) band intensity and peak position allows us to quantify the local Au oxide layer thickness. The EC-TERS data provide experimental evidence for the existence and nm spatial distribution of at least two AuOx species with distinct coordination numbers on the active site. With this operando Raman nanoscopy approach, a wide range of materials and reaction conditions can be explored to locate and characterize nanoscale

active surface sites, for example, in electro- or photoactive materials during operation or in biological environments under physiological conditions. The insight into molecular-scale material (re)activity gained in this way will pave the way for bottom-up defect design and rational tuning of device properties.

## Results

### Electrochemical TERS approach to study Au defect oxidation.
Figure 1a shows a schematic of the EC-TERS mapping experiment. Through electrochemical potential control of the Au(111) electrode, we reversibly switch between defect oxidation ON and OFF states: at a sample potential, $E_{sample}$, of 1.45 V vs. Pd-H, water is split to generate AuOx selectively at defect sites of low overpotential while the Au(111) terrace sites remain oxide free (ON). At $E_{sample} = 1.1$ V vs. Pd-H, the electrode including defects is in its reduced, metallic $Au^0$ state (OFF). The Au EC-STM tip, i.e., the plasmonic Raman enhancer, is scanned across the surface within the diffraction-limited laser far-field focus spot (diameter of ca. 500 nm). We record EC-TER spectra as a function of tip position to generate operando reactivity maps with nanometer spatial chemical resolution, i.e., providing chemical contrast between $Au^0$ and AuOx as a function of applied potential and surface site. Details about the EC-TERS setup and experimental procedures can be found in the Methods section.

Figure 1b shows a cyclic voltammogram (CV) of Au(111) in 0.1 M $H_2SO_4$ recorded in the EC-TERS sample cell. The sharp order/disorder transition peak at ~ 1.1 V vs. Pd-H (Supplementary Fig. 1 and Note 1) and the pronounced peak ~ 1.55 V vs. Pd-H corresponding to the electro-oxidation of Au(111) terrace sites are characteristic for clean, well-defined Au(111) electrodes of high quality[21]. The shoulder extending from roughly 1.32 to 1.48 V vs. Pd-H (green region) in the anodic scan direction toward more positive potentials prior to terrace and bulk oxidation is owing to water splitting at and selective oxidation of nanoscale surface defects[22,23]. The charge density under the defect oxidation peak amounts to ca. 60 $\mu C\,cm^{-2}$, in excellent agreement with literature[21]. In the cathodic scan toward more negative potentials, the peak from ~ 0.98 to 1.25 V vs. Pd-H (gray region) is owing to the reduction of (previously anodically formed) AuOx.

Figure 1c shows example EC-TER spectra of the ON (green) and OFF (black) states. When defect-catalyzed water splitting is ON, we detect a large band at ~ 560–580 $cm^{-1}$ that we assign to AuOx following previous experimental and theoretical literature assignments[24,25] (see Supplementary Note 2 and Fig. 2 for full-range spectra). In contrast, in the OFF state, no Raman peak is visible because all AuOx has been reduced. We can reversibly switch between ON and OFF states, in this way following in situ the activation, reaction and recovery of the Au electrocatalyst. The tip potential, $E_{tip}$, is always maintained at 1.0 V vs. Pd-H to ensure that the tip remains oxide free (see Supplementary Fig. 3 for a tip CV). As a result of the fixed tip potential, the tip-sample bias differs between ON and OFF, which might result in a slightly smaller tip-sample distance and thus higher TERS sensitivity in the OFF state (cf. Supplementary Note 3 for discussion). Such an effect was shown to be prominent for in air, but negligible for in-water TERS experiments[26]. The defect AuOx stretch mode is not observed in conventional Raman spectra (TERS probe retracted several μm from the surface, Supplementary Fig. 4 and Note 4), highlighting the strong near-field created in the tip-sample gap when the tip is approached in TERS conditions.

### Nanoscale chemical reactivity mapping.
Figure 2 shows correlated images of the EC-STM apparent topography (left a, c, e) and EC-TERS AuOx band intensity (right b, d, f) of the Au(111) electrode. For details about the contrast determination and

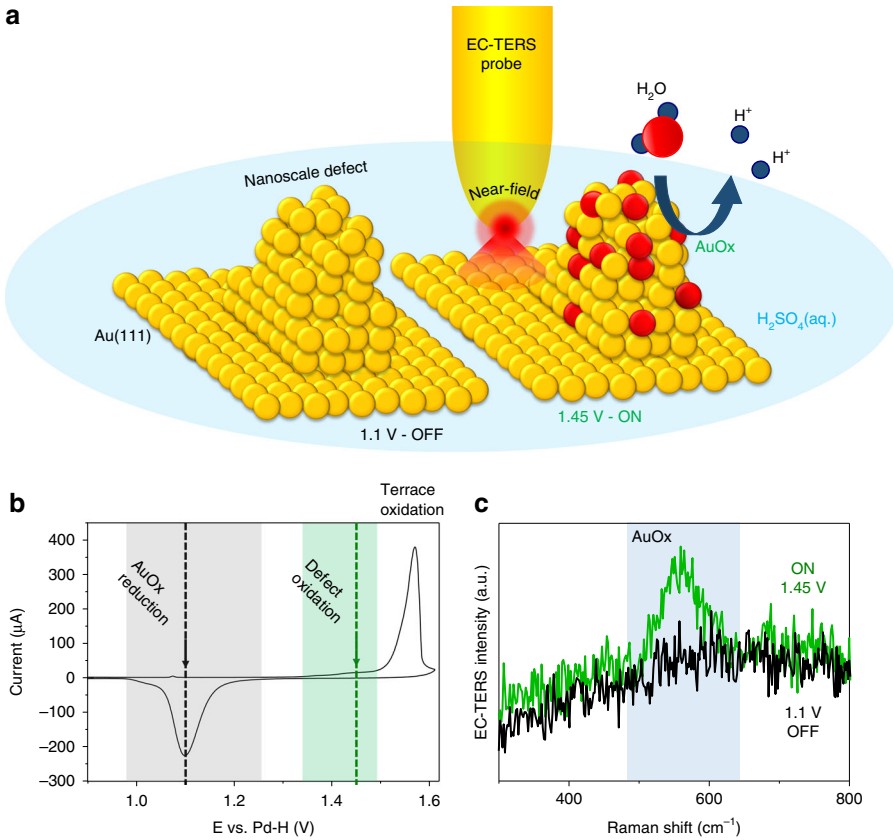

**Fig. 1 EC-TERS of selective and reversible Au nanodefect oxidation. a** Schematic of the EC-TERS operando nanoscopy approach: Defect oxidation OFF (left, 1.1 V vs. Pd-H) or ON (1.45 V vs. Pd-H) states can be spatially and chemically resolved by mapping the active site of interest with the EC-TERS probe with 9.4 nm spatial precision during electro-activation. **b** Cyclic voltammogram of Au(111) in 0.1 M $H_2SO_4$ recorded in the EC-TERS sample cell (scan rate: 50 mV s$^{-1}$). Green and gray areas highlight the defect oxidation and gold oxide (AuOx) reduction regions, respectively. **c** EC-TER spectra for ON (with AuOx peak at ca. 580 cm$^{-1}$) and OFF states recorded at the active (defect) site.

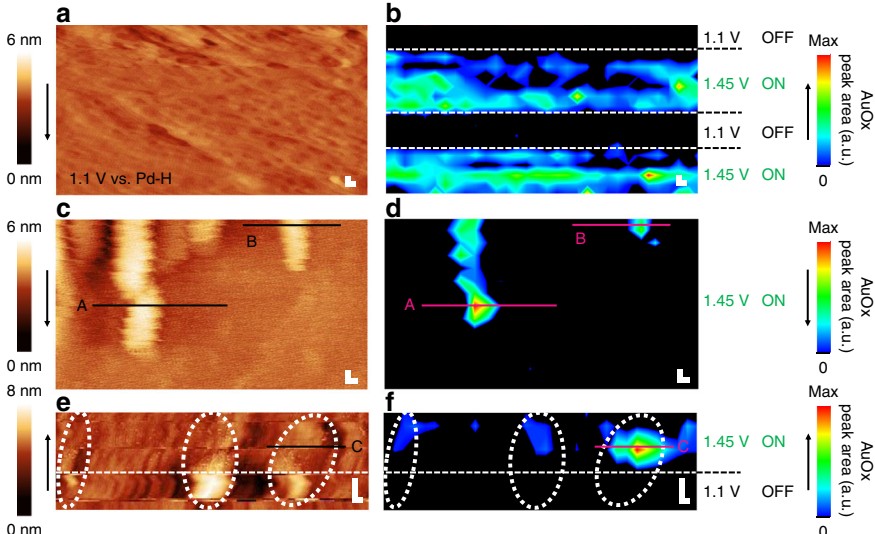

**Fig. 2 Operando 2D chemical imaging of catalytic defect reactivity. a**, **c**, **e** EC-STM images of Au(111) in 0.1 M $H_2SO_4$. Scan speed: 1 line s$^{-1}$. **b**, **d**, **f** Corresponding EC-TERS maps reconstructed from the gold oxide (AuOx) band intensity. Pixel size: 9.4 nm, 3 s **b**, **f** or 1 s **d** spectral acquisition time/pixel. **a**–**f** Tunneling current ($I_t$) = 1 nA, $E_{tip}$ = 1.0 V vs. Pd-H, $E_{sample}$ as indicated; scale bars = 10 nm. Black arrows indicate scan directions. Dotted white lines are guides to the eye. Solid black and pink lines are line profiles analyzed in Fig. 3.

EC-TERS image construction by linear background subtraction and peak area integration, see Supplementary Figs. 5–7 and Note 5. Figure 2a shows the electrode in the OFF state, electrochemically roughened by complete surface oxidation (at 1.6 V vs. Pd-H, Supplementary Fig. 8) and subsequent reduction cycles. During subsequent potential switching between OFF and ON, the operando EC-TERS map (Fig. 2b) visualizes the reversible water splitting and Au defect oxidation at good spectral contrast. Interestingly, the EC-TERS band intensity varies strongly between different locations during the ON scans, indicating a spatially heterogeneous defect density and/or activity on the roughened Au (111) surface on the 10 nm length scale.

To visualize the correlation between nanoscale defect structure and reactivity, we map nanodefects of ca. 20–40 nm width and 2–4 nm height present on the Au(111) single crystal. The apparent structural contours in the EC-STM map (Fig. 2c) are excellently reproduced in the simultaneously obtained chemical-contrast EC-TERS map (Fig. 2d), which suggests a chemical spatial resolution of at least the pixel size of 9.4 nm, to be discussed further below. Surprisingly, the catalytic activity of the topographically rather similar Au structures differs drastically: only two of the nanodefects visible in the EC-STM image exhibit clear AuOx EC-TER signals, whereas the others remain spectrally silent. We attribute these spectral differences to the local variations in surface charge and work function[27] owing to atomic active-site heterogeneities[28] that lead to the varying degrees of defect reactivity directly visualized by EC-TERS mapping.

Figure 2e, f demonstrate the full operando nanoscopy capability of EC-TERS where we image the activation of the nanodefect while switching from OFF (bottom) to ON (top) states halfway during the scan. In the OFF state, the EC-STM image indicates three distinctive structures of ca. 25 to 50 nm width and 2–4 nm height (Fig. 2e, white dotted lines); as expected,

the EC-TERS maps show no AuOx formation, neither at the nanodefects nor on the Au(111) terrace sites (Fig. 2f). When defect oxidation is initiated, the EC-STM noise level increases, similarly to what has been previously reported for hydrogen evolution on Pt[11]. The change in tip-sample bias and increased noise level due to Au oxidation lead to a significantly reduced STM image contrast, whereas the activated catalyst sites appear in the EC-TERS image because the simultaneously recorded EC-TER spectra exhibit the strongest AuOx peak intensities at the nanodefects. The flat gold region remains largely AuOx free, i.e., it does not exhibit significant water splitting reactivity.

**Correlating surface structure and chemical reactivity.** Furthermore, our approach allows us to correlate the apparent structure and chemical reactivity of the nanodefects by investigating the EC-STM and EC-TERS line profiles (solid lines marked in Fig. 2). The AuOx peak intensity (after background correction to exclude possible near-field artefacts, see Supplementary Note 6 for details) uniformly follows the changes in the EC-STM height profiles (Fig. 3a to c and Supplementary Figs. 9, 10). The apparent height of the defect steps is in the order of 0.5 nm, or of multiples of 0.5 nm, which corresponds excellently to the height of a monolayer AuOx of 0.47–0.5 nm as formed during complete oxidation of a Au monolayer and subsequent Au-O place exchange[29,30]. Changes in apparent topography are accompanied by changes in EC-TERS band intensity within typically one pixel, an effect particularly evident for the broad stepped structure profiled in Fig. 3c. These results underline that we resolve chemical heterogeneities with a spatial sensitivity of ~ 10 nm. Note that the step size was chosen as a compromise between measurement time and instrument stability; extrapolating from in-air TERS results where spatial resolution of 3 nm or better has been repeatedly

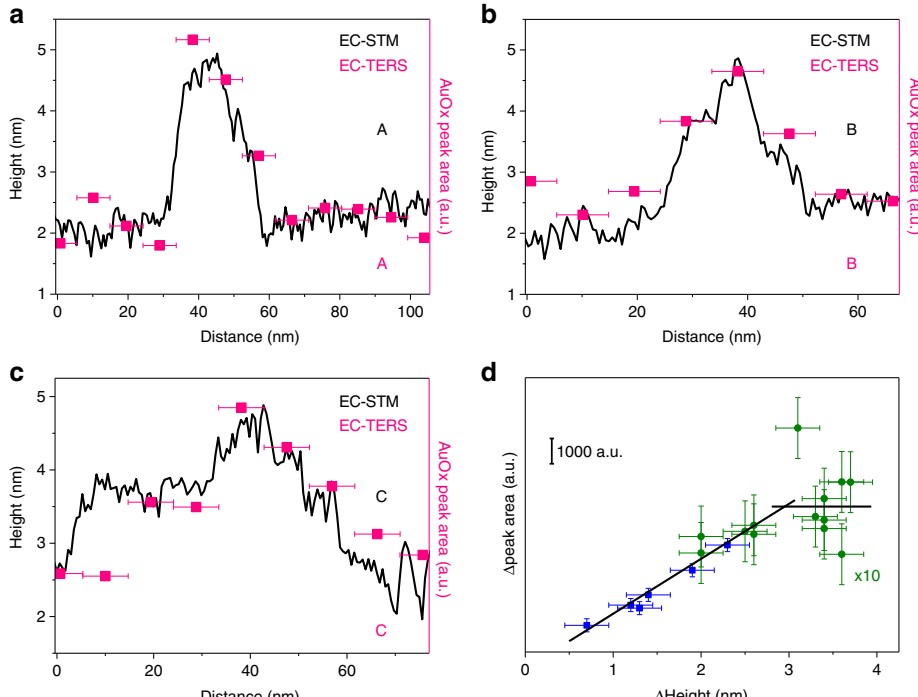

**Fig. 3 Correlation between EC-STM height profiles and EC-TERS band intensities. a–c** EC-STM (black) and corresponding EC-TERS (pink) line profiles from regions marked with A, B, C in Fig. 2c–f. The distance error bars (x-error) are estimated to be of the pixel size, i.e. ± 4.7 nm. **d** Difference in gold oxide (AuOx) peak area vs. difference in defect structure height. Blue and green data sets were recorded with different tips on different days and samples. The Δheight error bars are estimated to be ± 0.25 nm. The Δpeak area error bars are determined from the standard deviation either at flat Au regions or in the OFF state. Solid lines are guides to the eye.

demonstrated, it can be expected that improving the setup stability further against thermal drift and noise will enable comparable EC-TERS spatial resolution in the order of a few nanometers.

Plotting the AuOx peak area topographic height difference on the nanodefect displays a linear relation up to ~ 3 nm height difference before saturation of the EC-TERS signal (Fig. 3d). As the EC-TER signal intensity is linearly proportional to the number of scatterers, the observed linear increase in signal intensity with increasing amount of AuOx at the defects is reasonable. The intensity saturation above 3 nm structure height is likely due to a limited film thickness of AuOx. Surface AuOx growth was reported to be limited to three monolayer equivalents of oxygen, or 6 monolayer equivalents of AuOx owing to Au-O place exchange, and higher anodic potentials are required for Au oxidation of lower-lying (bulk) Au layers[21,29]. In other words, in the ON state, only surface oxidation of defects is selectively triggered up to a depth of maximum six monolayers of AuOx, corresponding to a AuOx film thickness of ca. 3 nm. We do not expect a significant lowering in plasmonic enhancement with increasing dielectric layer thickness > 2 nm[31]. For example, $Al_2O_3$ thin films of comparable thickness are commonly used as protective coatings on TERS tips and have been reported to not significantly alter the plasmonic enhancement[32]. Moreover, the feedback regulation of the EC-STM leads to a decreased tip-sample distance to compensate for a lowering of tunneling efficiency on AuOx compared to on Au at a fixed $I_t$, which in turn leads to an increased gap enhancement[26].

Our experimental approach allows us to directly correlate the single active-site EC-TERS nanoscopic information with the macroscopic CV that displays the average response of the whole catalyst electrode surface. Given the total electrode defect charge density of ca. $60 \, \mu C \, cm^{-2}$ and the surface-limited oxide growth of 6 ML AuOx at defect sites, we quantify the surface defect coverage to be about 4.5% for the well-defined Au(111) electrode, a non-negligible amount despite the high quality of the employed single crystal (cf. Supplementary Note 1 for derivation). Our data thus provides direct experimental evidence for the importance of few active sites for the total catalytic surface reactivity.

**Elucidating local Au nanodefect chemistry.** In addition to nanoscopic localization and quantification of reaction products, operando EC-TERS nanoscopy provides valuable insight into the local chemical nature of the surface specimen from band position analysis, owing to the extreme energy (frequency) resolution of Raman spectroscopy, here of $\sim 1.6 \, cm^{-1}$ (0.2 meV)[16]. Figure 4b shows an EC-TERS map reconstructed from the AuOx peak positions as obtained from an unconstrained single Gaussian peak fit (see Fig. 4a and Supplementary Note 6 for details) from the data set shown in Fig. 2f, ON. The AuOx peak position differs markedly between ca. 580 and $550 \, cm^{-1}$ as a function of probe location on the nanodefects. Notably, for relative flat areas on the defect, the peak is located at higher Raman shifts above the intermediate value of $565 \, cm^{-1}$ (Fig. 4b, c sides), whereas for stepped defect parts, the Raman shift is smaller than $565 \, cm^{-1}$ (Fig. 4c middle). Thus, we conclude that at least two distinct AuOx species of different geometric coordination are locally present within the nanodefects in accordance with earlier suggestions by the Weaver group[24]. For a tentative assignment of the two observed frequency ranges and circumstantial chemical identification of the formed AuOx species, we refer to previous literature results. It is known that, generally, the Raman shift decreases with a lowering of the metal-site coordination[20]. For (bulk) AuOx species, the two most prominent geometries, $Au_2O_3$ and $Au_2O$, were calculated to exhibit Au-O vibrations at 581 and

$563 \, cm^{-1}$, respectively[25], which are very close to the Raman shifts observed in our experiments. Thus, we speculate that on flatter defect-terrace-like areas, $Au_2O_3$ is generated, while sharper protrusions on the nanodefects favor $Au_2O$ formation (Fig. 4d). Such intra-defect spatial reaction heterogeneity has recently been predicted theoretically for electrocatalytic $H_2$ dissociation on Pt[1] but not yet been observed in situ experimentally. There are a lot of efforts being made in the electrochemical surface science community to achieve atomistic insights into electrode chemistry, in terms of both, experiment and theory. However, including dynamic charges and local fields in, for example, density functional or molecular dynamics simulations to obtain atomistic insights and valuable guidance for the interpretation of experimental data are far from trivial, and an increasing number of groups are actively pursuing this challenge worldwide. On the experimental side, in situ X-Ray experiments may provide a deeper insight into the exact nature of the AuOx species formed at nanodefects, although to date one would still have to bear the compromise of moving away from operando conditions. In this sense, the EC-TERS data at hand provides a more detailed contribution to the identification of the nature of the AuOx species generated at the active sites, which can be viewed as reaction intermediates, and may aid to clarify the reaction mechanism of electrocatalytic water splitting and of oxygen evolution at more positive potentials.

## Discussion

In conclusion, we have demonstrated how label-free operando EC-TERS nanoscopy can be employed in the field of corrosion and electrocatalysis to image inter- and intra-defect reactivity heterogeneity of water splitting on Au with $\sim 10 \, nm$ chemical spatial sensitivity. Starting from a pristine Au(111) catalyst surface without any pre-defined self-assembled organic monolayer attached, we have been able to perform EC-TERS 2D mapping under reaction conditions and detected reaction products, namely oxidized Au defects, on the nanoscale. The EC-TERS maps revealed site-specific surface oxidation of Au nanodefects up to a maximum of 3 nm AuOx layer thickness. The highly local spectral fingerprints indicate formation of at least two AuOx species with distinct coordination numbers located at defect-terrace or protrusion active sites that are tentatively assigned to $Au_2O_3$ and $Au_2O$, respectively. Correlating electrode averaged and nanoscopic site-specific information, we infer a defect coverage of ca. 4.5% on the high-quality Au(111) single crystal. Our results can be expected to aid, for example, the long-standing quests concerning the broad and asymmetric AuOx peak shape and chemical interpretation thereof[24] and the nanoscale distribution and chemical composition of reaction intermediates and products, essential for the development of advanced water splitting or oxygen evolution catalysts[33,34].

To move on from the described model oxidation reaction, a next logical step would be to probe a full catalytic reaction, reaching potentials of the oxygen evolution reaction (OER). However, there are several experimental challenges associated with EC-TERS of OER: we have not yet produced $O_2$ at potentials significantly higher than 1.6 V vs. Pd-H because this would require large bias potentials between tip and electrode to ensure that the EC-STM tip (EC-TERS probe) is not oxidized and remains oxide free. The effects of large bias potentials on TERS data, however, are not trivial to understand, for example, in terms of tip-sample distance control or field effects on the plasmonic gap behavior, rendering data interpretation much more complex[26]. Electrode materials with lower OER overpotentials, such as Pt or Pd, would be advantageous in that they allow for a smaller tip/sample bias but form much less efficient plasmonic

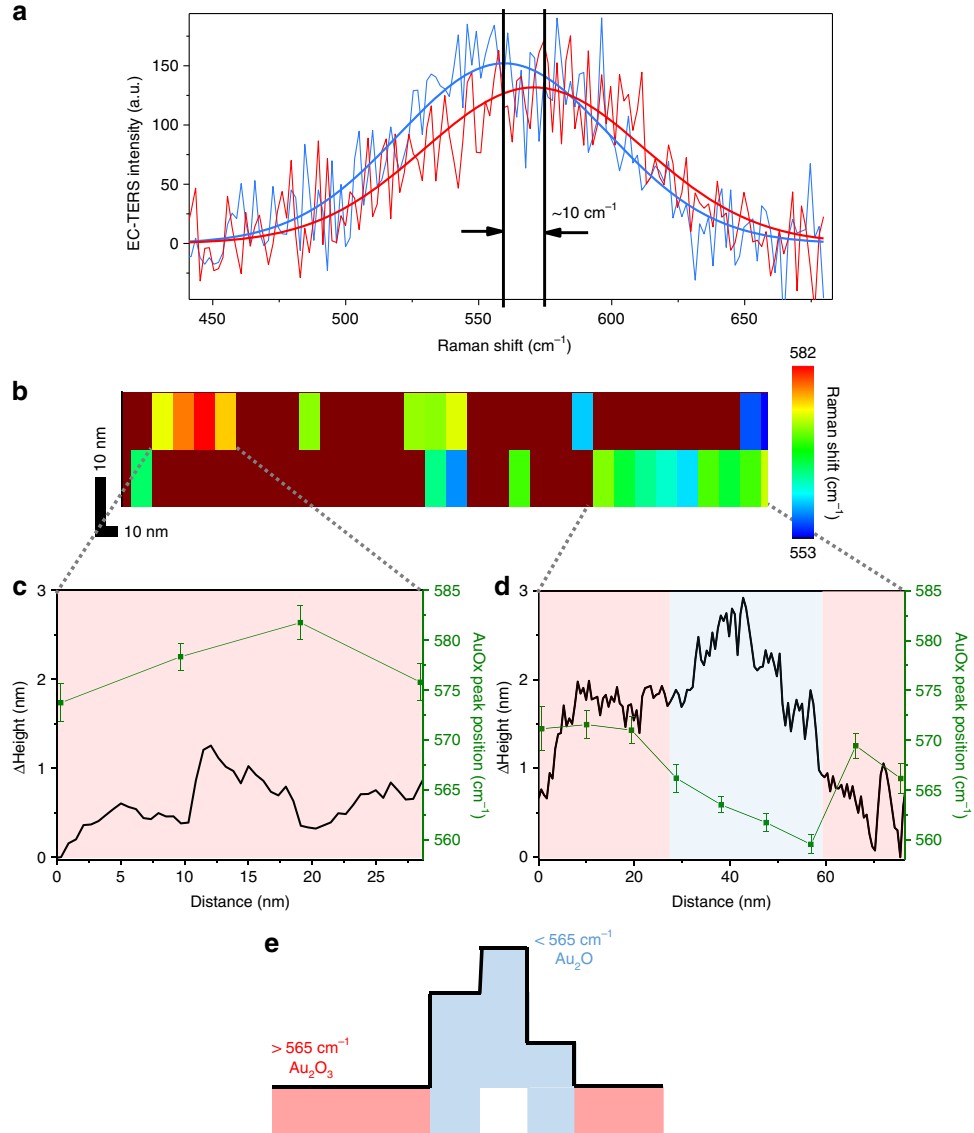

**Fig. 4 Nm-resolved chemical nature of AuOx species. a** Two example EC-TER spectra (background subtracted) recorded at different AuOx defect locations and corresponding Gaussian peak fits, exhibiting a difference in peak position of ~ 10 cm$^{-1}$. **b** EC-TERS map reconstructed from the AuOx peak position corresponding to the upper part of Fig. 2f. **c, d** Correlation between Δheight profiles and EC-TERS peak positions. The y-error bars represent the standard deviation of the bootstrapping fitting analysis (see Supplementary Note 6 for details). **e** Schematic illustrating the difference in AuOx peak position, i.e, in chemical nature, at different locations on the nanodefect.

gaps[35]. The detection sensitivity of the instrument would have to be significantly improved, for example, by installing a more sensitive CCD camera and improving the coupling of the laser focus to the plasmonic gap[36].

Moreover, the evolution of oxygen would introduce additional noise and instability in the STM mode, possibly adding artefacts to the TER spectra/images (as well as to the STM images). It is unclear how the presence of the tip in close vicinity to the electrode might affect the flow of oxygen bubbles and thus the local reactivity underneath the tip in the nanometer-sized near-field region. One might consider to develop a horizontal TERS setup with an inclined tip configuration reminiscent of the setups described in ref. [14,37] to avoid steric (tip) hindrance of oxygen (bubble) diffusion.

Despite these technical challenges, the EC-TERS imaging approach holds the promise to be employed on a wide range of systems beyond electrocatalytic materials where nano-site activity determines macroscopic device behavior: elucidating nucleation and growth mechanisms of functional materials at surfaces, investigating membrane channels or biomimetic trigger functions in bio-chemical surfaces or exploring dopant centers or domain boundaries in 2D optoelectronic materials with operando Raman nanoscopy will aid to push our understanding of the driving forces for chemical and energy conversion to the molecular level.

## Methods

**EC-TERS setup**. The employed EC-TERS setup is a home-built device[38]. In brief, the setup uses an Olympus 50× long working-distance objective (working distance = 10.6 mm, NA = 0.5). The red HeNe laser (632.8 nm, linearly polarized, maximum output power of 35 mW) is focused onto the electrochemically etched Au-tip of an electrochemical STM (5420, Keysight Technologies GmbH). The focusing/collection objective is arranged in a side-illumination configuration with 55° angle between the objective and the substrate surface normal. The Raman signal is detected with a Horiba iHR 550 spectrograph with a nitrogen-cooled CCD camera (Symphony II, Horiba). A 600 lines/mm grating was used for the reported experiments. The setup covers a wide energy range from 156 cm$^{-1}$ (cutoff of the employed dichroic filter) to 4000 cm$^{-1}$, thus including low lying phonon modes of advanced electrode materials such as oxides or chalcogenides and interactions

between organic or inorganic adsorbates and metals, the "fingerprint region" of organic species (and their interactions with electrolyte molecules) as well as high-frequency modes of water and similar modes.

**Electrode materials and preparation procedures**. The Au(111) single crystal (MaTecK, 10 mm diameter) was flame-annealed under Ar (6 N, Westfalen) atmosphere. The aqueous electrolyte was 0.1 M $H_2SO_4$ (96%, Merck, Suprapur) prepared from MilliQ water (Millipore-Q, 18 MΩ cm) or deuterated water (Deuterium oxide, 99.9 atom % D, Sigma-Aldrich). A Pt (0.5 mm diameter, Alfa Aesar, Premion, 99.997% metals basis) or Au wire (0.5 mm diameter, Alfa Aesar, Premion, 99.9985% metals basis) was used as counter electrode and freshly flame-annealed and rinsed with MilliQ water before use. The reference electrode was a hydrogen-loaded Pd wire (Pd-H). All potentials in this manuscript are reported versus Pd-H. The Pd wire (0.5 mm diameter, MaTecK, 99.95% metals basis) was filled with hydrogen by immersing it into 0.1 M $H_2SO_4$ and applying 5–10 V between the Pd wire and the Au counter electrode until the hydrogen evolution corresponded roughly to the oxygen gas bubble evolution. All glassware used in this study was boiled in 40% $HNO_3$ (≥ 65%, Sigma-Aldrich) and subsequently rinsed and boiled three times in MilliQ water. The Au-tips were electrochemically etched in a 1:1 mixture of HCl (37% fuming, Merck, Emsure)/Ethanol (Merck, Emsure). A voltage of 2.4 V was applied between the Au-tip wire (0.25 mm diameter, Alfa Aesar, Premion, 99.9985% metals basis) and the counter electrode (Au wire of 1 mm diameter shaped to a ring). The Au-tips were electrically isolated using Zapon wax.

**Spectral acquisition parameters**. The employed laser power of ~ 6–8 mW power at the focus spot was determined in air and estimated to be lowered to ~ 1–3 mW in electrolyte due to optical aberrations at the glass window. The spectral acquisition time ranged between 1 and 5 s and is specified at the appropriate sections. The data analysis was performed with matlab.

## Data availability

All relevant source data are available from the corresponding author upon request.

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

## Acknowledgements

We thank Mischa Bonn, Juan M. Feliu, Bogdan Marekha, Natalia Martín Sabanés, Sapun Parekh, Dmitry Turchinovich, and Ulmas E. Zhumaev for valuable scientific discussions and comments on the manuscript. J.H.K.P. and K.F.D. gratefully acknowledge financial support by the Max Planck Graduate Center with the Johannes Gutenberg University Mainz (MPGC). K.F.D. acknowledges generous support through the Emmy Noether Program of the Deutsche Forschungsgemeinschaft (DO1691/1-1) and through the "Plus 3" Program of the Boehringer Ingelheim Foundation.

## Author contributions

K.F.D. and J.H.K.P. conceived the experiments and analysis procedures. J.H.K.P. conducted the experiments and performed the data analysis. M.B. and G.G. supported the mapping experiments and contributed to manuscript preparation. K.F.D. supervised the project. J.H.K.P. and K.F.D. wrote the manuscript. All authors discussed the results and commented on the manuscript.

## Competing interests

The authors declare no competing interests.
