## [Peer Review File · Nature Communications]

Reviewers' comments:

Reviewer #1 (Remarks to the Author):

The paper by Domke et al shows a simultaneous EC-STM and EC-TERS study of the initial stages of gold surface oxidation. This is a significant experimental advance, showing the spatially heterogeneous surface oxidation and the potential observation of two different oxides at the same potential.

My only comment concerns the very loose definition of "electrocatalysis" used by the authors. In fact, there is no electrocatalysis in the paper, as not a single molecule of O₂ is produced, and in fact gold is a bad catalyst for OER. The authors should not frame this as "an important showcase heterogeneous catalytic reaction" as gold surface oxidation is not catalysis and it is not that important... What the authors study, are the (heterogeneity effects on the) initial stages of gold surface electro-oxidation. They should say so in the paper and in the title of the paper, and refrain from statements that they show that EC-TERS nanoscopy can be used for imaging electrocatalysis, because in fact this is not what they show. A similar statement applies for the authors' definition of "water splitting", when in fact they mean (again) surface oxidation. Of course they can make an outlook statement about applications for electrocatalysis. I would be curious to know what happens to the TERS signal when oxygen is truly evolving (though this happens at much higher potential) - did the authors actually ever go that positive in potential?

Reviewer #2 (Remarks to the Author):

This manuscript by Pfisterer et al. reports on mapping chemistry of AuO_x formation on defect-sites of Au(111) using an EC-STM based TERS approach. In this work, the authors use electrochemical control to regulate water splitting chemistry on Au surface to form AuO_x complexes. The AuO_x complexes are determined using their Raman signatures which is locally detected by the STM-tip with a sub-10 nm chemical-spatial resolution.

The key concept of "Chemical site-specificity on the nanoscale" that the authors have based their work on has definitely been one of the main objectives of TERS since its inception. However, a similar report considering this key concept has previously been reported (J. Phys. Chem. Lett. 2019, 10, 11, 2817-2822) and describes site-specific chemistry using TERS at the liquid-solid interface. While there are some excellent aspects to their work, based on the novelty I don't believe this article is suitable for publication in Nature Communications.

Notwithstanding, following comments might be of use to the authors:

1. Figure 3 and Figure S9 shows that the TERS cross-sectional line profiles follow a similar trend to its corresponding STM cross-sections. A direct-correlation between the EC-TERS and EC-STM cross-sectional profiles seems too good to be true. For example, if a particular site is more active on the defect structure that is anywhere other than the topmost part of the structure, then TERS map would show that site as the highest point in the TERS cross-sectional profile.
2. The authors subtract the background for all their TERS images. Mapping TERS-background vs. TERS AuO_x peaks can provide valuable information in elucidating the chemical hot-spots vs. plasmonic hot-spots.
3. The amount of laser intensity at the sample position (~ 1- 3 mW) appears to be somewhat high. From personal experience, even in liquid conditions, 100-200 μW is sufficient to observe TERS. Nonetheless, it is surprising that in this report the TERS probe only acts as a point detector and rather doesn't induce any chemistry.
4. While I believe that the experiment is extremely challenging, the quality of TERS maps are not of the highest quality. Also, to claim that pixel limited (sub-10 nm) resolution based off of one pixel in a TERS image is a little bit difficult to take in.

Reviewer #3 (Remarks to the Author):

This manuscript describes electrochemical tip-enhanced Raman spectroscopy for combined topography and reactivity imaging of electroactive surface sites under realistic reaction conditions. The electrochemical water splitting on Au nano-defects is examined and the formation of different gold oxides on different sites is observed. The presented work is of exceptional quality and really showcases the amazing possibilities of TERS.

Besides enthusiasm and a recommendation to publish this work in Nature Communications, I don't have any significant comments for improvement besides a few remarks given below:

Comments:

- Abstract:

can be expected to be a game changer \diamond is an expected game changer

nano-defects as showcase energy conversion \diamond nano-defects, a showcase energy conversion

- Page 5:

The authors say: "The shoulder extending from roughly 1.32 to 1.48 V vs. Pd-H (green region) in the anodic scan direction toward more positive potentials prior to terrace and bulk oxidation is due to water splitting at and selective oxidation of nanoscale surface defects" but they then comment that part of the Raman signal observed at 1.45 V can come from terrace-like sites. Does terrace oxidation also start before 1.48V? The "prior to terrace and bulk oxidation" might be misleading here.

- Page 7:

The text here is a bit misleading because it first states: "Figure 2 shows correlated images of the EC-STM apparent topography (left A,C,E) and ECTERS AuOx band intensity (right B,D,F) of the electrochemically roughened Au(111) electrode" but then says later on that Figure 2C is a Au(111) single crystal. It is not clear for a person outside the field that the roughened electrode and the single crystal electrode are the same electrode, so one could think that the sample shown in Figure 2A is a different sample than the sample on Figure 2C.

Response to Reviewers

Reviewer #1:

The paper by Domke et al shows a simultaneous EC-STM and EC-TERS study of the initial stages of gold surface oxidation. This is a significant experimental advance, showing the spatially heterogeneous surface oxidation and the potential observation of two different oxides at the same potential.

My only comment concerns the very loose definition of "electrocatalysis" used by the authors. In fact, there is no electrocatalysis in the paper, as not a single molecule of O₂ is produced, and in fact gold is a bad catalyst for OER. The authors should not frame this as "an important showcase heterogeneous catalytic reaction" as gold surface oxidation is not catalysis and it is not that important.

What the authors study, are the (heterogeneity effects on the) initial stages of gold surface electro-oxidation. They should say so in the paper and in the title of the paper, and refrain from statements that they show that EC-TERS nanoscopy can be used for imaging electrocatalysis, because in fact this is not what they show.

A similar statement applies for the authors' definition of "water splitting", when in fact they mean (again) surface oxidation.

Of course they can make an outlook statement about applications for electrocatalysis.

Response: We agree with Reviewer 1 that people might be confused by the terminology in the manuscript concerning "electrocatalysis" and "water splitting" in combination with the oxidation of gold surfaces.

While, in fact, water is split at the Au electrode at very specific potentials to form gold oxide and to release protons into the electrolyte - which is what we referred to as electrochemical water splitting and surface oxidation -, we agree that from a classical point of view, the oxidation of gold surfaces is not viewed as a catalytic reaction, but rather as a model corrosion reaction as no oxygen is continuously evolved from the surface. We would like to reiterate that, in our manuscript, we do show that EC-TERS nanoscopy is capable to detect reaction products, namely AuO_x, during an electrochemical reaction.

The behavior of gold single crystal electrodes in aqueous electrolytes and the pathways of oxidation of gold surfaces represent important model mechanistic processes in electrochemistry that continue to lead to important understanding in corrosion phenomena and (electro)catalytic reactions [e.g. Koper and co-workers in Chemical Science 4 (2013) 2334 or PCCP 16 (2014) 13583; Guo et al. Nature Comm. 7 (2016) 1348].

To avoid confusion, we have rephrased the title of the manuscript and replaced "electro-catalytic" and added the "Au defect oxidation": "Nanoscale reactivity mapping of Au defect oxidation under electrochemical reaction conditions"

Furthermore, we have adjusted the wording throughout the manuscript to prevent misconception of our results, as detailed in the following:

Page 1, Abstract:

"We map the electrochemical oxidation of Au nano-defects, a showcase energy conversion and corrosion reaction, with a chemical spatial sensitivity of <10 nm."

Page 3, 2nd paragraph:

"As a showcase corrosion and heterogeneous catalysis-related reaction²⁰, we image the oxidation of nanoscale protrusions at a Au(111) single crystal electrode as resulting from electrochemical water splitting at defect sites."

Page 4, Figure 1: The term 'water splitting' is removed to not potentially confuse readers.

"Figure 1 | EC-TERS of selective and reversible Au nanodefekt oxidation. (A) Schematic of the EC-TERS *operando* nanoscopy approach: Defect oxidation OFF (left, 1.1 V vs. Pd-H) or ON (1.45 V vs. Pd-H) states can be spatially and chemically resolved by mapping the active site of interest with the EC-TERS probe with 9.4 nm spatial precision during electro-activation."

Page 5, 1st paragraph: We have replaced the term "water splitting" by "defect oxidation".

Page 13, 2nd paragraph:

“In conclusion, we have demonstrated how label-free *operando* EC-TERS nanoscopy can be employed to image inter- and intra-defect reactivity heterogeneity of **electrochemical** water splitting **and Au oxidation** with <10 nm chemical spatial sensitivity.”

Page 14, 2nd paragraph: “As a brief outlook, the EC-TERS imaging approach can be employed on a wide range of systems beyond **electro-catalytic materials** where nano-site activity determines macroscopic device behavior: ...”

I would be curious to know what happens to the TERS signal when oxygen is truly evolving (though this happens at much higher potential) - did the authors actually ever go that positive in potential?

Response: This is a very exciting question that we would like to investigate, too. To date, we have not yet produced O₂ at potentials significantly higher than 1.6 V vs. SHE because this would require large bias potentials between tip and electrode in order to make sure that the EC-STM tip or EC-TERS probe is not oxidized and remains oxide free. The effects of large bias potentials on the TERS data, however, are not trivial to understand, for example, in terms of tip-sample distance control or field effects on the plasmonic gap behavior, rendering data interpretation much more complex. Electrode materials with lower OER overpotentials (allowing for a smaller tip/sample bias), such as Pt or Pd, would be advantageous but form much less efficient plasmonic gaps, i.e. the detection sensitivity is insufficient in our current setup. Moreover, the evolution of oxygen would introduce additional noise and instability in the STM mode, again adding artefacts to the TER spectra/images (as well as to the STM images) that are not yet fully understood. Also, it is unclear how the presence of the tip in close vicinity to the electrode might affect the flow of oxygen bubbles and thus the local reactivity underneath the tip in the nearfield/ probed nanometer region. One might consider to develop a “horizontal” TERS setup with an inclined tip configuration to avoid steric (tip) hindrance of oxygen (bubble) diffusion.

In view of these technical difficulties, so far we have avoided strong O₂-gas evolution and currently focus on monitoring other types of electrocatalytic and –synthetic surface reactions.

Reviewer #2:

This manuscript by Pfisterer et al. reports on mapping chemistry of AuOx formation on defect-sites of Au(111) using an EC-STM based TERS approach. In this work, the authors use electrochemical control to regulate water splitting chemistry on Au surface to form AuOx complexes. The AuOx complexes are determined using their Raman signatures which is locally detected by the STM-tip with a sub-10 nm chemical-spatial resolution. The key concept of “Chemical site-specificity on the nanoscale” that the authors have based their work on has definitely been one of the main objectives of TERS since its inception.

However, a similar report considering this key concept has previously been reported (J. Phys. Chem. Lett. 2019, 10, 11, 2817-2822) and describes site-specific chemistry using TERS at the liquid-solid interface.

While there are some excellent aspects to their work, based on the novelty I don’t believe this article is suitable for publication in Nature Communications.

Response: The search for a nano-tool that provides means to perform *operando* monitoring of electrochemical reactions on the nanometer scale with adjusted kinetics and tuning of the Fermi level (by changing the applied electrode potential) is one key feature for the identification of active sites, elucidation of local reaction mechanisms and of site-specific local (defect-)chemistry on the nanometer scale under reaction conditions, ideally combined in a single experimental approach – and one of the long-standing dreams in surface science that had not yet been achieved.

Indeed, the recent publication of Bhattarai and El-Khoury (J. Phys. Chem. Lett. 2019, 10, 11, 2817-2822) images the dimerization reaction of a p-nitrothiophenol (NTP) to dimercaptoazobenzene with AFM-based TERS in a droplet of water (now added as a reference on page 2/3). While this work might appear to some extent similar, a closer inspection reveals essential differences crucial for establishing EC-TERS in the field of *operando* surface science, e.g. in electro-catalysis/-synthesis or corrosion science:

From the technical point of view, the Bhattarai/El-Khoury study similarity to our work is that it provides a chemical image of a solid-liquid interface with sub-10 nm spatial chemical resolution. It does however, not provide any control over the chemical state of the surface as opposed to our electrochemical potential-control approach. As such, with our approach, *we are able to initiate, reverse and/or stop the chemical conversion at the interface at will*, comparable to T/p-control in classical UHV surface science approaches, thus providing *operando* conditions. Even more interestingly, the electrochemical approach allows to control the electrode potential in such a way that the electrochemical oxidation reaction occurs *only at desired (defect) sites* (and not on flat Au terraces).

Related to the point of (chemical) control over the investigated system, Bhattarai/El-Khoury use the TERS hot spot both to generate and to probe the chemical conversion, rendering data interpretation in terms of mechanistic insights difficult. Note that, surprisingly, they do not find a spatial correlation between the hot-spot activity and plasmon-triggered conversion, and also not between reactant and product distribution (temporal and spatial). Our experimental EC-TERS approach, on the other hand, is designed to *avoid tip-induced artefacts, separating (electro)chemical reactivity from plasmon-based interrogation*. Similarly, our STM-based EC-TERS allows us to precisely control the tip-potential and to keep the Au tip oxide free, in this way eliminating another potential source for spectral artefacts compared to the AFM-based liquid TERS approach.

From the chemistry point of view, we provide *correlated nanoscale topography and EC-TERS images, both in the 'reactant' and 'product' states, and, most importantly, also during the transition*. This unprecedented combination of data allows us to elucidate the relation between local surface chemistry and atomic-scale topography. Bhattarai/El-Khoury also speculate that the observed spatial heterogeneity in their experiment might be due to local surface structure, but they do not provide (AFM) evidence. As such, the effect of surface structure on the chemical dimerization mechanism of NTP and the resulting product (e.g. *cis* or *trans* form) remains unresolved in their paper, in contrast to our evidence for spatially heterogeneous formation of different oxide species dependent on the nanodefekt roughness.

In addition to correlating nanoscale topography with nanoscale chemistry, and in contrast to the Bhattarai/El-Khoury study, our work also discusses *the relation between nanoscopic and macroscopic (ensemble) electrode behavior*. We provide direct experimental evidence that few (electro)active defect sites are responsible for the macroscopically observed electrode reactivity, as postulated regularly in the field of surface science. Mechanistic understanding of how nanosite-specific activity influences macroscopic device (e.g. reactor, large-scale catalyst, etc.) reactivity in terms of product selectivity, conversion rate, yield etc. can be expected to have a huge implication on the design of reactive interfaces.

Furthermore, all EC-TERS (8 reports) and liquid TERS studies (i.e. without potential control; 10 reports) have focused their efforts on pre-defined self-assembled organic monolayers that were prepared before the actual TERS experiments. Our work, for the first time, investigates surface behavior *starting from a 'clean' metal surface*, i.e. we follow the formation of Au oxide, an inorganic species, during the water splitting reaction, not pre-assembled beforehand. For the fields of (electro)catalysis and corrosion science, we believe that it is absolutely crucial to be able to start with a pristine metal/catalyst surface that is then modified throughout the reaction, for example by surface oxidation, or a continuously evolving reaction – as it is the case in realistic operation settings, i.e. we mimic actual *operando* conditions.

In summary, our work clearly goes beyond what has been demonstrated so far, both in terms of experimental capability as well as in scientific elucidation of defect behavior. We show that EC-TERS provides the means to simultaneously acquire topographic as well as chemical information on the 10 nm-level under oxidation reaction conditions splitting water and forming gold oxide. As such, a multiplicity of research questions on the

nanoscale level in the fields of fundamental electrochemistry, bioelectrochemistry, corrosion science, electrocatalysis, catalysis-related energy conversion, electrosynthesis and other (electrified) solid-liquid phase related disciplines can now be addressed experimentally.

Notwithstanding, following comments might be of use to the authors:

1. Figure 3 and Figure S9 shows that the TERS cross-sectional line profiles follow a similar trend to its corresponding STM cross-sections. A direct-correlation between the EC-TERS and EC-STM cross-sectional profiles seems too good to be true. For example, if a particular site is more active on the defect structure that is anywhere other than the topmost part of the structure, then TERS map would show that site as the highest point in the TERS cross-sectional profile.

Response: Indeed, our data shows a direct correlation between EC-TERS and EC-STM cross-sectional profiles. The EC-TERS intensity is directly correlated to the number of detected scatterers, i.e. amount of gold oxide formed. This means that if more gold oxide was formed, for example, at the side of the nano-defect, we would expect the side region to exhibit the highest EC-TERS intensity. However, this is not what we observe experimentally for the particular system under investigation. As we discuss in the manuscript (Figure 3D), depending on the height of the nanostructural defect, we see a correlated EC-TERS intensity that saturates at about 3 nm. This is due to the limited Au oxide growth at the nanodefekt and has been suggested previously in the literature from macroscopic CV experiments on Au electrodes [Jerkiewicz et al., Limit to extent of formation of the quasi-two-dimensional oxide state on Au electrodes. *J. Electroanal. Chem.* **422**, 149–159 (1997)]. The highest EC-TERS intensity at the nanodefekts thus originates from the increased number of AuOx scatterers below the EC-TERS probe due to higher gold defect oxidation reactivity. As discussed in the manuscript, our findings are in agreement with the literature portraying that defects are oxidized at 1.45 V vs. Pd-H, while the flat Au(111) terraces remain oxide free and confirm macroscopic experimental findings in the literature [Zhumaev et al., Electro-oxidation of Au(1 1 1) in contact with aqueous electrolytes: New insight from in situ vibration spectroscopy. *Electrochim. Acta* **112**, 853–863 (2013)]. As such, in summary, the topography-chemistry correlation is not as surprising as it may seem at a first glance.

2. The authors subtract the background for all their TERS images. Mapping TERS-background vs. TERS AuOx peaks can provide valuable information in elucidating the chemical hot-spots vs. plasmonic hot-spots.

Response: Yes, certainly the TERS background contains valuable information which we always investigate in our work. We would like to draw the attention to Figure S6E in the SI that shows a non-background subtracted EC-TERS reactivity map. The fact that background-subtracted and non-background-subtracted EC-TERS maps (Figure 2 versus Figure S6E) are essentially the same indicates that the background does not significantly change as a function of surface location. A constant background, in turn, indicates that the plasmonic tip-sample gap remains constant, as stated in the SI Page 3, 2nd paragraph:

“Note that the spectral background does not change when switching between ON and OFF states, indicating that the EC-TERS gap resonance is unaffected by the potential switch.”

3. The amount of laser intensity at the sample position (~1- 3 mW) appears to be somewhat high. From personal experience, even in liquid conditions, 100-200 μW is sufficient to observe TERS. Nonetheless, it is surprising that in this report the TERS probe only acts as a point detector and rather doesn't induce any chemistry.

Response: The laser intensity of 1 to 3 mW is higher than for in-air experiments because of the typically distorted laser far-field focus and thus inefficient laser-plasmon coupling due to light aberrations at the air/glass/electrolyte interface [see e.g. Martín Sabanés et al., A versatile side-illumination geometry for tip-enhanced Raman spectroscopy at solid/liquid interfaces. *Anal. Chem.* **88**, 7108–7114 (2016)]. Also, aqueous electrolytes are known to act as excellent heat sink for EC-TERS experiments compared to in-air experiments.

It is not possible to provide a general statement as to the optimal power (and integration time) for TERS experiments. The required laser power depends on the (EC-)TERS setup (electrochemical cell versus liquid drop,

side illumination versus bottom illumination, collection pathway, optical alignment, detection sensitivity of camera, excitation wavelength and match with gap mode and/or electronic states, tip quality etc.) and on the system studied (inorganic versus organic, scattering cross section of investigated modes, number of scatterers in nearfield, nearfield damping effects of molecules/species in the gap (local dielectric function), etc.). In general, our excitation power for EC-TERS experiments varies between 100 μ W and 3 mW, depending mostly on the system under study, the quality of the tip and the alignment precision of the optical pathway. We typically choose lower powers (and longer times and/or more spectra for averaging) for more delicate, organic adlayers that burn easily. Here, for the mapping experiment of an inorganic AuOx layer, we found the given parameters (relatively short integration times and higher powers) to provide the best imaging results in terms of contrast and stability.

Regarding plasmon-induced (likely heat-induced) artefacts, we do not find any indication for illumination-induced chemistry (see excellent temporal stability of EC-TER spectral signature shown in Figure S5). EC-surface-enhanced Raman spectroscopy (EC-SERS) experiments on polycrystalline Au electrodes in liquid electrolytes analogous to our work have used significantly higher laser powers of 50 to 70 mW without triggering additional (electro)chemical reactivity [e.g. Desilvestro and Weaver, Surface structural changes during oxidation of gold electrodes in aqueous media as detected using surface-enhanced Raman spectroscopy, *J. Electroanal. Chem.*, 1986, 209, 377–386.]. The electrode potentials could possibly be slightly shifted according to the Nernst equation and a local heating effect. However, such potential shifts have not been observed in the mentioned EC-SERS studies nor by us.

4. While I believe that the experiment is extremely challenging, the quality of TERS maps are not of the highest quality. Also, to claim that pixel limited (sub-10 nm) resolution based off of one pixel in a TERS image is a little bit difficult to take in.

Response: TERS imaging under electrochemical *operando* conditions is indeed a very challenging task. TERS in air or under ultra-high vacuum conditions and even TERS in liquid or EC-TERS under *non-reaction* conditions is very different from EC-TERS experiments under *operando* conditions in that the EC system under investigation is typically out of equilibrium (potential controlled) which can lead to current fluctuations, and even subtle thermal fluctuations, electric noise, instabilities in the electric contacts etc can lead to changes in the chemical state of the surface that will be reflected in the spectroscopic and voltammetric data in terms of unspecific temporal fluctuations. As such, a direct comparison of map quality to published non-EC work is misleading. For example, regarding the acquisition of optimal EC-TERS and EC-STM images, a huge compromise is required in terms of tip shape: a tip to provide sufficient EC-TERS signal is ideally cone shaped and thus deviates from an ideal, monoatomically sharp EC-STM tip. EC-STM tunneling instabilities often occur, and particularly for long measurement times during a mapping experiment, utmost care has to be taken to exclude noise-related artefacts. Aside from the ITO-study by the Van Duyne group with significantly lower spatial resolution, no 2D EC-TERS images have been published till date despite the fact that a handful of groups are working on this task, clearly showing the enormous challenges associate with this experiment.

Concerning our statement of sub-10 nm chemical spatial sensitivity, the sharp rise in EC-TERS intensity across the nanodeflects suggests that our actual chemical spatial feature sensitivity is even lower than the pixel size of 9.4 nm. In the TERS community, the chemical spatial resolution is often determined by going from 0 to 90 % TERS intensity as a function of distance, which in our case is within one pixel of 9.4 nm. As such, it is safe to conclude that the spatial chemical sensitivity is not worse than the step size. Note that we have deliberately refrained from speaking about “optical resolution” as we have not determined the spatial resolution in the sense the term is used in the field of microscopy but rather speak about “chemical resolution”, “feature sensitivity” or “spatial sensitivity” in the manuscript.

Reviewer #3:

This manuscript describes electrochemical tip-enhanced Raman spectroscopy for combined topography and reactivity imaging of electroactive surface sites under realistic reaction conditions. The electrochemical water splitting on Au nano-defects is examined and the formation of different gold oxides on different sites is observed. The presented work is of exceptional quality and really showcases the amazing possibilities of TERS.

Besides enthusiasm and a recommendation to publish this work in Nature Communications, I don't have any significant comments for improvement besides a few remarks given below:

Response: We thank Reviewer 3 for the very positive feedback about our work and for sharing with us the enthusiasm about the great possibilities that EC-TERS now has opened up.

Comments:

- Abstract:

**can be expected to be a game changer \diamond is an expected game changer
nano-defects as showcase energy conversion \diamond nano-defects, a showcase energy conversion**

Response: We have changed the wording according to the suggestions of Reviewer 3.

- Page 5:

The authors say: "The shoulder extending from roughly 1.32 to 1.48 V vs. Pd-H (green region) in the anodic scan direction toward more positive potentials prior to terrace and bulk oxidation is due to water splitting at and selective oxidation of nanoscale surface defects" but they then comment that part of the Raman signal observed at 1.45 V can come from terrace-like sites. Does terrace oxidation also start before 1.48V? The "prior to terrace and bulk oxidation" might be misleading here.

Response: Terrace oxidation starts only at potentials more positive than 1.48 V vs. Pd-H, i.e. only defects are selectively oxidized at 1.45 V vs. Pd-H while flat Au(111) terraces are kept oxide free. However, on the nanodeflect structure itself, we observe and discriminate between more flat "defect-terrace-like" and more protrusion-like defect structures. To avoid confusion, we have adapted and rephrased according to the comments of Reviewer 3:

Page 1, Abstract:

The results indicate the reversible, concurrent formation of spatially separated Au₂O₃ and AuO₂ species at **defect-terrace** and protrusion sites on the defect, respectively.

Page 14, 1st paragraph:

The highly local spectral fingerprints indicate formation of at least two AuOx species with distinct coordination numbers located at **defect-terrace** or protrusion active sites that are tentatively assigned to Au₂O₃ and Au₂O, respectively.

- Page 7:

The text here is a bit misleading because it first states: "Figure 2 shows correlated images of the EC-STM apparent topography (left A,C,E) and ETERS AuOx band intensity (right B,D,F) of the electrochemically roughened Au(111) electrode" but then says later on that Figure 2C is a Au(111) single crystal. It is not clear for a person outside the field that the roughened electrode and the single crystal electrode are the same electrode, so one could think that the sample shown in Figure 2A is a different sample than the sample on Figure 2C.

Response: We have rephrased the paragraph accordingly:

"Figure 2 shows correlated images of the EC-STM apparent topography (left A,C,E) and EC-TERS AuOx band intensity (right B,D,F) of the **Au(111)** electrode. For details about the contrast determination and EC-TERS image construction by linear background subtraction and peak area integration, see Supplementary Materials Figures S6 to S8. Figure 2A shows the electrode in the OFF state, **electrochemically roughened by complete surface oxidation (at 1.6 V vs. Pd-H) and subsequent reduction cycles.**"

Reviewers' comments:

Reviewer #1 (Remarks to the Author):

I would like to see some of the discussion in the rebuttal about the (technical) difficulties of looking at real catalysts and real catalytic conditions in the revised paper. Now the narrative of the paper still suggests that the authors are close doing catalysis, while this is actually no so obvious or trivial.

Reviewer #2 (Remarks to the Author):

I agree with the authors about the novelties of this paper but the paper allures more towards the electrochemical community rather than for the plasmonics and spectroscopy community. The following comments reflect my insights regarding the paper:

1. Spatial resolution in STM based TERS with sub-9nm resolution is not surprising and not convincingly supported with the data. I disagree with the authors that this is the convention that is used to determine the spatial resolution. In order to make a stronger stance on the resolution aspect of this work, a much finer step sizes would be required to see the overall rise in the TERS signal, if any.
2. What governs the TERS image? This again boils down to the step sizes. Does the localized field govern your TERS intensities?
3. This paper lacks description of very fundamentals that the plasmonics and TERS communities strive for. Inclusions regarding the plasmonics of the tip, sample, and the tip-sample junction can strongly bolster the scope of this paper.
4. The assignments of the AuOx Raman peak is not convincing. It can be notoriously difficult to assign a Raman spectra from a single peak.

Response to Reviewers

Reviewer #1:

I would like to see some of the discussion in the rebuttal about the (technical) difficulties of looking at real catalysts and real catalytic conditions in the revised paper. Now the narrative of the paper still suggests that the authors are close doing catalysis, while this is actually not so obvious or trivial.

Response: We have included an additional paragraph in the outlook section that discusses the challenges of monitoring OER and other catalytic systems with EC-TERS.

“To move on from the described model oxidation reaction, a next logical step would be to probe a full catalytic reaction, reaching potentials of the oxygen evolution reaction (OER). However, there are several experimental challenges associated with EC-TERS of OER: we have not yet produced O₂ at potentials significantly higher than 1.6 V vs. Pd-H because this would require large bias potentials between tip and electrode to ensure that the EC-STM tip (EC-TERS probe) is not oxidized and remains oxide free. The effects of large bias potentials on TERS data, however, are not trivial to understand, for example, in terms of tip-sample distance control or field effects on the plasmonic gap behavior, rendering data interpretation much more complex²⁶. Electrode materials with lower OER overpotentials, such as Pt or Pd, would be advantageous in that they allow for a smaller tip/sample bias but form much less efficient plasmonic gaps³⁵. The detection sensitivity of the instrument would have to be significantly improved, for example, by installing a more sensitive CCD camera and improving the coupling of the laser focus to the plasmonic gap³⁶. Moreover, the evolution of oxygen would introduce additional noise and instability in the STM mode, possibly adding artefacts to the TER spectra/images (as well as to the STM images). It is unclear how the presence of the tip in close vicinity to the electrode might affect the flow of oxygen bubbles and thus the local reactivity underneath the tip in the nanometer-sized near-field region. One might consider to develop a “horizontal” TERS setup with an inclined tip configuration reminiscent of the setups described in Ref.^{14,37} to avoid steric (tip) hindrance of oxygen (bubble) diffusion. Despite of these technical challenges, the EC-TERS imaging approach holds the promise to be employed on a wide range of systems beyond electro-catalytic materials...”

Reviewer #2:

I agree with the authors about the novelties of this paper but the paper allures more towards the electrochemical community rather than for the plasmonics and spectroscopy community. The following comments reflect my insights regarding the paper:

1. Spatial resolution in STM based TERS with sub-9nm resolution is not surprising and not convincingly supported with the data. I disagree with the authors that this is the convention that is used to determine the spatial resolution. In order to make a stronger stance on the resolution aspect of this work, a much finer step sizes would be required to see the overall rise in the TERS signal, if any.

Response: Indeed, it would be desirable to perform EC-TERS imaging experiments with smaller step sizes. Unfortunately, this is currently not possible with our setup. Essentially, the choice of step size boils down to a compromise between STM stability, or drift, and pixel integration time, or detection sensitivity. We have tried to use smaller step sizes, albeit at the expense of signal/to noise at lower integration times that then have to be employed to stay out of the range of STM-tip drift (i.e. to keep the total time to acquire one image the same). A new PhD student is now working on setup improvements that will allow for smaller step sizes, but as any technical modification on the setup requires extensive testing, this will likely take a year before we obtain reliable, reproducible data considering smaller step sizes. Note that current state-of-the-art TERS imaging of a potential-controlled electrode surface (and the only example so far) by the Van Duyne group¹⁹ reaches ca. 80 nm resolution. We want to emphasize again that the EC approach cannot be directly compared with in-air, in-UHV or in-liquid instruments, see last response to reviewers.

The topic of the possibility of achieving smaller step sizes, i.e. better spatial chemical resolution was addressed in the main paper text on p10, centre:

“Note that the step size was chosen as a compromise between measurement time and instrument stability; extrapolating from in-air TERS results where spatial resolution of 3 nm or better has been repeatedly demonstrated, it can be expected that improving the setup stability further against thermal drift and noise will enable comparable EC-TERS spatial resolution in the order of a few nm.”

Additionally, the issue of thermal drift and its effect on the EC-TERS / EC-STM image comparison was addressed in the SI, p8.

In the original manuscript it reads (and our data Fig 3 A-C and Fig S9 show) that “Changes in apparent topography are accompanied by changes in EC-TERS band intensity within typically one pixel”. As such, following the conventionally accepted approach in TERS imaging [e.g. Zhang et al. Nature 2013 “Chemical mapping of a single molecule by plasmon-enhanced Raman scattering”, Zhong et al. Nature nanotechnology 2017 “Probing the electronic and catalytic properties of a bimetallic surface with 3 nm resolution”, Chen et al. Nature communications 2014 “A 1.7nm resolution chemical analysis of carbon nanotubes by tip-enhanced Raman imaging in the ambient”], we deduce that our step size is the resolution-limiting factor here.

2. What governs the TERS image? This again boils down to the step sizes. Does the localized field govern your TERS intensities?

Response: (Added to the SI) “In general, Raman intensities are governed by the local field strength – for TERS, this corresponds to the local field enhancement, or near-field strength, in the tip-sample gap – and by the amount of scatterers for a given experimental configuration, detection sensitivity and scattering cross section of the system under study. The effect of the formation of a plasmonic tip-sample gap on the field (enhancement) can be seen (albeit not quantified as there is no signal in the conventional Raman spectrum) from the comparison of the conventional (no tip-sample gap) and TERS signals as shown in S4.

For self-assembled monolayers of organic molecules adsorbed at metal substrates, sometimes a local increase in TERS intensity between a factor five to ten at step edges or surface protrusions compared to TERS at neighbouring flat Au regions, partly also accompanied by frequency shifts of Raman modes, has been reported in the literature¹⁰⁻¹⁵. Such edge effects can be attributed to local heterogeneities in the plasmonic properties of the formed tip-sample gap that differ depending on the actual (atomic) gap geometry and field localisation and polarisation. As a result, the LSP or gap resonance shows nanoscale spatial heterogeneities both in intensity and in location of the resonance maximum causing differences in coupling efficiency between excitation far-field laser and gap mode. Strongest near-fields are created under resonance conditions when the excitation wavelength approximately matches the LSP resonance maximum. Therefore, shifts in the gap plasmonic resonance maximum (intensity and position) can lead to significant differences in TER scattering intensities. Furthermore, highly localised strong fields can lead to a local Stark effect that causes shifts in vibrational frequencies¹². Also different adsorption geometries of molecules at step edge sites compared to terrace sites can account for differences in TER shifts¹³.

In general, the plasmonic (gap) properties can be deduced from the shape (plasmon resonance energy) and the intensity (field strength, or enhancement) of the spectral background¹⁶.

Figure S2 shows a comparison between TER signals recorded in ON and OFF conditions. Evidently, the background does not differ, neither in shape nor in intensity. As discussed on p6 in the main text of the manuscript, changes in the tip-sample distances and thus in the field enhancement upon variation of the tip-sample bias (due to the variation in applied potential), have previously been shown to be negligible in water,²⁶ in line with the conclusion drawn from Figure S2 about the unaltered field enhancement in air compared to in water. As such, it is fair to assume that the field strength and enhancement do not change upon potential change and that the obvious change in peak intensity is due to a change in the amount of scatterers present in the nearfield, i.e. the amount of AuOx formed.

Furthermore, Figure S10 shows raw data from an example line scan. From these spectra it is evident that the TERS background recorded at oxide-free terrace sites and the background recorded on the oxidized defect are identical, i.e. there is no measurable difference between the plasmon resonance energy or field enhancement above terrace or defect sites. As such, we have attributed the oxide intensities recorded on defect sites and their spatial variation across defects to a variation in the local number of AuOx scatterers, or the thickness of the formed defect oxide layer.

Regarding the spatial uniformity of the background signal, we also refer to Fig S6E. Here, the EC-TERS map without background subtraction (field enhancement and number of scatterers determine intensity) is plotted – and is essentially the same as the one presented in the main text in Figure 2D after background correction to present a “pure” chemical map corrected for (however small) near-field spatial variations (number of scatterers determine change in peak intensity). Comparing Figs S6E and 2D, it is evident that, independent of whether with or without background correction, the quantitative result of selective AuOx formation on the defects remains the same.

In summary, while edge effects due to spatially heterogeneous plasmonic gap properties were observed in TERS imaging of self-assembled monolayers, our data does not provide any indication for measurable differences in tip-sample coupling neither as a function of tip position nor of applied potential within the employed potential range. Therefore, we attribute the TER intensity differences to spatial- and/or potential-dependent variations in the amount of AuOx scatterers.”

3. This paper lacks description of very fundamentals that the plasmonics and TERS communities strive for. Inclusions regarding the plasmonics of the tip, sample, and the tip-sample junction can strongly bolster the scope of this paper.

Response: We have added one page with a more detailed description of the fundamentals of including selected literature that provides a thorough introduction to the most important aspects of TERS to the SI to provide more background about the technique to the interested non-specialist.

“TERS is based on the excitation of localised surface plasmons (LSPs) in the apex of a metal (or metallized) SPM tip. In our configuration, we employ a commercially available electrochemical scanning tunnelling microscope

(EC-STM) coupled to a Raman optical platform as described in Materials and Methods in the main text. The working principle of TERS is based on the efficient coupling of a focused laser beam with the tip to excite LSPs, which generates a strong electromagnetic near-field at the very apex of the STM tip. In a way, the tip acts as an antenna and converts, or “concentrates”, far-field radiation into a nm-confined field underneath the tip. The near-field character of the Raman excitation provides the extreme spatial optical resolution, typically of a few nm for in-air experiments, which depends essentially on the size, curvature and surface (atomic) topography of the tip apex.

The created near-field typically is a factor 10 to 100 larger than the excitation far-field and can induce Raman scattering in the species or molecules located at nm-close distance below the STM tip. The Raman scattering intensity, in a first approximation, scales with the fourth power of the magnitude of the excitation field. The strong field enhancement at the tip compared to the far-field intensity enables the detection of very few surface scatterers down to single molecules. When the substrate is a metal or exhibits metal optical properties similar to the ones of the tip, LSP excitation in the tip is accompanied by a corresponding image dipole formation in the substrate which leads to even higher field enhancements in the tip-sample gap. Accordingly, such a TERS configuration is called gap-mode TERS and provides typically even higher sensitivity than TERS on a dielectric substrate.

The interested reader is referred to recent reviews about TERS, its fundamentals and applications^{8,9}.

4. The assignment of the AuOx Raman peak is not convincing. It can be notoriously difficult to assign a Raman spectrum from a single peak.

Response: Yes, it is indeed notoriously difficult to assign (EC-)TERS (or, in general, Raman/IR vibrational) modes. As such, we rely on previous band assignments in literature where AuOx formation has been intensively studied with vibrational spectroscopies and by simulations on the ensemble level. Regarding the identification of AuOx species, we carefully phrased in the main text on p13 centre: “Thus we speculate that [...] Au₂O₃ is generated, while sharper protrusions [...] favour Au₂O ...”

For the case of the typically observed broad 560-580 cm⁻¹ mode, previous assignments – based on ensemble vibrational responses – were always made to “various oxide species” [25,26] (cf. main text p13, here specific reference to the Weaver group). Our EC-TERS approach has the advantage that it provides an extremely local probe of only 10 nm. Interestingly, we find that the peak position varies as a function of local position on the nanodefekt (main text p13) – an observation that has not yet been made because of the lack of suitable tools. There are a lot of efforts being made in the electrochemical surface science community to achieve atomistic insights into electrode chemistry, in terms of both, experiment and theory. However, including dynamic charges and local fields in, for example, DFT or MD simulations to obtain some guidance for the interpretation of our (and of ensemble) spectra is far from trivial, though worldwide a few groups are actively pursuing this challenge. On the experimental side, *in-situ* X-Ray experiments may provide a deeper insight into the nature of the AuOx species formed at nanodefekts, although one would have to bear the compromise of moving away from *operando* conditions. In any case, the necessity to develop novel *operando* tools that provide chemical insight into reactive electrode nano-sites is recognized by the respective communities – which is also our driving force for the EC-TERS imaging development and the work at hand – and we are sure that there will be continuous progress to achieve this goal.

To account for the uncertainty in band assignment, we have re-phrased the respective sentences in the main text in the following ways:

P6 top: “When defect-catalyzed water splitting is ON, we detect a large band at around 560 to 580 cm⁻¹ that we assign to AuOx following previous experimental and theoretical literature assignments^{24,25} (see Fig. S2 for full-range spectra).”

P13 centre: “...in accordance with earlier suggestions by the Weaver group²⁴. For a tentative assignment of the two observed frequency ranges and circumstantial chemical identification of the formed AuOx species, we refer to previous literature results.”

P13 centre/bottom (added): “There are a lot of efforts being made in the electrochemical surface science community to achieve atomistic insights into electrode chemistry, in terms of both, experiment and theory. However, including dynamic charges and local fields in, for example, density functional or molecular dynamics simulations to obtain atomistic insights and guidance essential for the interpretation of experimental data is far from trivial, and an increasing number of groups are actively pursuing this challenge worldwide. On the experimental side, *in-situ* X-Ray experiments may provide a deeper insight into the exact nature of the AuO_x species formed at nanodefects, although to date one would still have to bear the compromise of moving away from *operando* conditions. In this sense, the EC-TERS data at hand provides a new, more detailed contribution to the identification of the nature of the AuO_x species [...] and may aid to clarify...”